# The Effect of Ametropia on Glaucomatous Visual Field Loss

**DOI:** 10.3390/jcm10132796

**Published:** 2021-06-25

**Authors:** Eun Young Choi, Raymond C. S. Wong, Thuzar Thein, Louis R. Pasquale, Lucy Q. Shen, Mengyu Wang, Dian Li, Qingying Jin, Hui Wang, Neda Baniasadi, Michael V. Boland, Siamak Yousefi, Sarah R. Wellik, Carlos G. De Moraes, Jonathan S. Myers, Peter J. Bex, Tobias Elze

**Affiliations:** 1Schepens Eye Research Institute of Massachusetts Eye and Ear, Boston, MA 02114, USA; eunyoung.choi@duke.edu (E.Y.C.); ray_wong@meei.harvard.edu (R.C.S.W.); Thuzar.thein@mah.harvard.edu (T.T.); mengyu_wang@meei.harvard.edu (M.W.); dianli@ds.dfci.harvard.edu (D.L.); jljinqy@jlu.edu.cn (Q.J.); 108014@jlufe.edu.cn (H.W.); nbaniasa@bidmc.harvard.edu (N.B.); 2Department of Ophthalmology, Harvard Medical School, Boston, MA 02114, USA; 3Department of Ophthalmology, Duke University Medical Center, Durham, NC 27705, USA; 4Department of Ophthalmology, Icahn School of Medicine at Mount Sinai, New York, NY 10029, USA; louis.pasquale@mssm.edu; 5Massachusetts Eye and Ear, Harvard Medical School, Boston, MA 02115, USA; lucy_shen@meei.harvard.edu; 6Department of Psychology, Jilin University, Changchun 130012, China; 7Jilin University of Finance and Economics, Changchun 130117, China; 8Wilmer Eye Institute, Johns Hopkins University School of Medicine, Baltimore, MD 21287, USA; michael_boland@meei.harvard.edu; 9Hamilton Eye Institute, University of Tennessee Health Science Center, Memphis, TN 38103, USA; siamak.yousefi@uthsc.edu; 10Bascom Palmer Eye Institute, University of Miami School of Medicine, Miami, FL 33136, USA; swellik@med.miami.edu; 11Edward S. Harkness Eye Institute, Columbia University, New York, NY 10032, USA; cvd2109@columbia.edu; 12Wills Eye Hospital, Thomas Jefferson University, Philadelphia, PA 19107, USA; jmyers@willseye.org; 13Department of Psychology, Northeastern University, Boston, MA 02115, USA; p.bex@neu.edu

**Keywords:** glaucoma, ametropia, myopia, hyperopia, visual field, OCT, SITA standard 24-2, pattern deviation, mean deviation, spherical equivalent

## Abstract

Myopia has been discussed as a risk factor for glaucoma. In this study, we characterized the relationship between ametropia and patterns of visual field (VF) loss in glaucoma. Reliable automated VFs (SITA Standard 24-2) of 120,019 eyes from 70,495 patients were selected from five academic institutions. The pattern deviation (PD) at each VF location was modeled by linear regression with ametropia (defined as spherical equivalent (SE) starting from extreme high myopia), mean deviation (MD), and their interaction (SE × MD) as regressors. Myopia was associated with decreased PD at the paracentral and temporal VF locations, whereas hyperopia was associated with decreased PD at the Bjerrum and nasal step locations. The severity of VF loss modulated the effect of ametropia: with decreasing MD and SE, paracentral/nasal step regions became more depressed and Bjerrum/temporal regions less depressed. Increasing degree of myopia was positively correlated with VF depression at four central points, and the correlation became stronger with increasing VF loss severity. With worsening VF loss, myopes have increased VF depressions at the paracentral and nasal step regions, while hyperopes have increased depressions at the Bjerrum and temporal locations. Clinicians should be aware of these effects of ametropia when interpreting VF loss.

## 1. Introduction

Glaucoma is an optic neuropathy characterized by progressive loss of retinal ganglion cells, resulting in optic nerve damage and eventual visual field (VF) loss. Since glaucoma tends to produce specific VF defects, the pattern deviation (PD) plot, which shows relative light sensitivity normalized by age-matched controls at each VF location, is crucial for the diagnosis of this optic neuropathy. Standard automated perimetry, particularly the Swedish interactive thresholding algorithm (SITA) standard 24-2 [1,2] is a widely used tool to characterize and monitor functional vision loss from glaucoma [3,4].

High myopia is considered a risk factor for glaucoma in several studies [5,6,7]. It is well-known that refractive error is associated with ocular biometric features. In general, myopic eyes tend to have a longer axial length and are more prolate than emmetropic eyes, while hyperopic eyes tend to have a shorter axial length and are more oblate (Figure 1A) [8,9]. In addition, the superior and inferior arcuate retinal nerve fiber bundles lie closer to the fovea in myopes compared to emmetropes or hyperopes (Figure 1B), resulting in a thicker temporal peripapillary retinal nerve fiber layer (RNFL) in myopic eyes [10,11,12,13]. Previous studies have shown the association between the spherical equivalent (SE) of refractive error and various anatomical parameters of the optic nerve head (ONH), which serve as important diagnostic criteria for glaucoma [12,14,15]. For example, increasing myopia is associated with greater optic disc torsion and tilt [14,15]. Furthermore, we have previously shown that the central retinal vessel trunks (CRVT), where retinal vessels enter and exit the optic disc, are located more nasally in myopes compared to hyperopes [15]. The nasalization of CRVTs, in turn, has been correlated with a central pattern of VF loss [16,17,18]. These findings suggest that myopes and hyperopes, with their varying structural parameters, may also have different patterns of light sensitivity. Previous works show myopia to be a risk factor for paracentral VF defects in glaucomatous eyes [19,20,21,22], while others report a high incidence of temporal VF defects in highly myopic eyes without known glaucoma [23]. We sought to build upon these studies by systematically examining the interaction effect of the full range of ametropia and VF loss severity on global VF patterns. Our goal is to understand how functional vision is affected by ametropia in patients with glaucoma.

In this study, we investigate the relationship between ametropia and VF patterns utilizing a large VF dataset from 5 academic institutions. Furthermore, we study the role of VF loss severity in modulating this relationship. We hypothesize that (A) given the structural differences in the eye, ametropia is associated with distinct patterns of light sensitivity, regardless of glaucoma; (B) because myopes have retinal nerve fiber (RNF) bundles that lie closer to the fovea, there is an interaction effect between glaucoma severity and ametropia; and (C) because myopes have more nasalized CRVTs, they develop deeper central VF depression (Figure 1). Our study aims to help clinicians better identify and interpret glaucomatous VF loss patterns in myopic and hyperopic patients.

## 2. Methods

The VFs used for this study were obtained through the Glaucoma Research Network, a multicenter consortium, which consists of Massachusetts Eye and Ear, Wilmer Eye Institute, New York Eye and Ear Infirmary, Wills Eye Hospital, and Bascom Palmer Eye Institute. The institutional review board of each participating institution approved this retrospective study. This study adheres to the Declaration of Helsinki and all federal and state laws.

### 2.1. Participants and Data

Our dataset consisted of SITA standard 24-2 VFs measured with the Humphrey Field Analyzer (HFA; Carl Zeiss Meditec, Dublin, CA, USA). The dataset used in this study consisted of all available VFs from the glaucoma services of Massachusetts Eye and Ear, Wilmer Eye Institute, New York Eye and Ear Infirmary, and Wills Eye Hospital, and the entire set of VF measurements from Bascom Palmer Eye Institute. The reliability criteria for VF selection were as follows: fixation losses ≤ 33%, false-negative rate ≤ 20%, and false-positive rate ≤ 20% [17,24,25,26,27]. If more than one measurement per eye fulfilled the reliability criteria, the most recent reliable VF was selected for each eye. VFs from the left eye were reflected along the vertical axis to match the orientation of the right eye, which is the standard orientation displayed in this paper. At testing time, the operator was required to enter the patient’s distance refractive error into the HFA machine in order for the machine to determine the matching trial lens. These distance refractive error values were logged by the HFA and used in the present study. The HFA device automatically assigns a value of 0 to all participants wearing a contact lens; therefore, all eyes with a distance refractive error of 0 could not be distinguished whether they were naturally emmetropic, pseudophakic, and emmetropic due to successful cataract surgery, or corrected by contact lenses and thus were excluded from analysis. In our supplemental analyses, additional exclusion criteria were applied based on age, SE, and mean deviation (MD): patients younger than 18 years or older than 80 years, eyes with −1.5 D ≤ SE ≤ +1.0 D, and eyes with MD less than −18 dB were excluded.

### 2.2. Statistical Analyses

All statistical analyses were performed using the R platform [28]. For patients with minimal VF loss, defined as MD within ±1 dB, mean PD and their standard deviations were plotted against SE for each VF location on the Humphrey 24-2. Linear regression slopes of PD and SE were calculated and plotted for patients with MD within ±1 dB and for those with MD < −12 dB. Furthermore, PD values at each VF location were modeled by linear regression with SE, MD, and their interaction (SE × MD) as regressors, using the following equation: PD~SE + MD + (SE × MD). Finally, given our previous finding that CRVT nasalization was associated with VF loss in the central 4 VF locations [17], SE slopes were calculated for the 4 most central locations, as a function of the magnitude of MD. *p* values of the slopes were adjusted for multiple comparisons by the false discovery rate method [29]. A *p* value < 0.05 was considered statistically significant.

## 3. Results

A total of 120,019 VFs from 120,019 eyes of 70,495 patients met our inclusion criteria. Figure 2 summarizes the clinical and demographic information of the subjects.

In the analysis involving all eyes, MD had a weak but statistically significant correlation with SE (Pearson’s r = 0.045, *p* < 2.2 × 10^−16^). Figure 3 shows the mean PD values at each of the 52 VF locations, grouped by bins of SE (bin centers: −6, −4, −2, 0, 2, 4, and 6 Diopters (Ds), bin width: ±1 D), for individuals with minimal VF loss (MD within ±1 dB). The following general trend was observed: with increasing myopia (decreasing SE), PD values increased at the peripheral VF locations and decreased at the central VF locations; opposite effects were noted for hyperopia.

Given the generally monotonic pattern of correlation observed, linear regression of PD from SE was performed to quantify the relationship. The regression coefficients at each VF location are shown in Figure 4. For patients with minimal VF loss (MD within ±1 dB), positive coefficients were observed in the paracentral and temporal VFs, indicating that increased myopia was associated with decreased light sensitivity in these regions. Negative slopes were observed mostly in the Bjerrum and nasal step areas, indicating that increasing hyperopia was associated with lower light sensitivity in these regions (Figure 4A). These results were in line with the trend observed in Figure 3. The significant positive slopes ranged from 0.01 to 0.04, and significant negative slopes ranged from −0.01 to −0.11 (*p* < 0.05). This means that for individuals with at most mild glaucoma, high myopes (SE: −6 D) can have up to 0.48 dB lower and 1.3 dB higher PD values compared to high hyperopes (SE: +6 D) at individual VF locations.

For patients with severe VF depression (MD < −12 dB), the pattern was slightly different: positive slopes were observed mostly in the paracentral VF, and negative slopes were observed in the Bjerrum and temporal regions (Figure 4B). This implies that increasing myopia was associated with VF depression in the paracentral region, and increasing hyperopia was associated with depression in Bjerrum and temporal regions. The magnitudes of the slopes were greater for severe VF loss compared to mild VF loss: the significant positive slopes ranged from 0.01 to 0.19, and significant negative slopes ranged from −0.02 to −0.23 (*p* < 0.05). This means that for severe glaucoma, high myopes (SE: −6 D) can have up to 2.3 dB lower and 2.8 dB higher PD values than high hyperopes (SE: +6 D) at individual VF locations.

To further explore the relationship between SE and PD, and to understand the role of VF loss severity on this correlation, linear regression was carried out with SE, MD, and their interaction term (SE × MD) as regressors. Figure 5A shows the “pure” SE effect on PD: when MD was not taken into account, myopes had a significantly lower light sensitivity in the paracentral and temporal VFs, but greater light sensitivity in the Bjerrum and nasal step regions. When the interactive effect of MD and SE was examined, myopic VF depression became localized to the paracentral and nasal step regions while hyperopic VF depression became more pronounced at the Bjerrum and temporal areas (Figure 5B). The significant positive interaction coefficients ranged from 0.002 to 0.012, and significant negative coefficients ranged from −0.002 to −0.01 (*p* < 0.05). The detailed regression coefficients for SE, MD, and SE × MD at each of the 52 VF locations are provided in Appendix A. As expected, the effect of MD alone on PD showed a highly significant correlation at every location.

Example VFs of myopic and hyperopic patients seen at Mass. Eye and Ear displaying these VF loss patterns are shown in Figure 6. With worsening glaucoma, myopic individuals tend to develop deeper paracentral VF defects, while hyperopic individuals tend to develop greater VF depression in the Bjerrum and temporal regions.

Finally, we examined the effect of SE on PD at the 4 most central VF locations (marked by the red squares in Figure 1C) as a function of VF loss severity. Myopia was significantly correlated with decreasing PD values at the central 4 locations (*p* < 0.001), and the correlation became stronger with decreasing MD (Table 1).

## 4. Discussion

In this study, we systematically investigated and quantified the effect of ametropia on retinal sensitivity at each VF location in the 24-2 pattern. While effects of myopia on specific VF defects have been reported [19,20,21,22,23], to our best knowledge, no prior work has examined this relationship in detail over the full range of ametropia and over the entire VF test locations. Additionally, we studied the interactive effect of ametropia and glaucoma severity on VF loss patterns. Our results show that while the effects of ametropia on individual PD values are small, there are distinct patterns of VF loss associated with myopia and hyperopia, and the relationship becomes stronger with increasing VF loss severity.

The Glaucoma Research Network dataset does not contain ophthalmic diagnoses, but given the origins of this large dataset, we may safely assume that VF loss occurring in these patients is mostly due to glaucoma. We first hypothesized that because of the structural variations in myopic and hyperopic eyes [8,9], there would be differences in light sensitivity depending on the degree of ametropia, regardless of the presence of VF loss. We demonstrate that in patients with minimal VF loss, patterns of light sensitivity differ among myopes and hyperopes, with myopes having relatively decreased sensitivity in the paracentral and temporal VF areas and hyperopes in the Bjerrum region and nasal step areas (Figure 4A). Notably, the different patterns in Figure 4A,B indicate a possibly independent effect of ametropia from that of nerve fiber anatomy associated with ametropia on VF loss. Therefore, we chose to statistically disentangle these two effects. Figure 5A shows the “pure SE effect”, i.e., the effect without accounting for the variance explained by VF loss severity. As expected, a pattern similar to Figure 4A is seen, with myopes having decreased sensitivity in the paracentral, inferior Bjerrum, and temporal areas. These “pure SE effects” could originate from ocular anatomical parameters associated with (axial) ametropia, but could also result from lens related diseases (e.g., nuclear cataract) or even by trial lens related measurement artifacts. Without medical diagnoses, potentially confounding diseases could not be controlled for in the current study. In a previous work on high myopia, Ohno-Matsui et al. [23]. carefully controlled for diseases and excluded trial lens artifacts by applying soft contact lenses for perimetry. They studied 492 highly myopic eyes without known glaucoma: among the eyes with significant VF defects, temporal field defects were observed in 61.5% of the eyes with round discs, 75.0% of the eyes with vertically oval discs, and 68.2% of the eyes with obliquely oval discs. Consistent with their results, our study found the temporal field to be the dominant location of reduced sensitivity in myopia. While they focus only on extremely myopic patients, we examine the full range of ametropia and show that myopic and hyperopic individuals, regardless of VF loss severity, have distinct patterns of light sensitivity.

We also hypothesized that, given the anatomical differences in RNF bundle trajectories between myopic and hyperopic eyes, there would be an interaction effect of ametropia and glaucoma severity on VF patterns. As mentioned above, using linear modeling with the interaction term (SE × MD) as a regressor, we were able to disentangle the effect of SE from that of MD. Lens artifacts and diseases of the anterior segment such as cataracts are most likely additive to VF loss patterns but would not interact with glaucomatous VF loss severity. This means, the “pure SE effect” bundles all possibly artifactual lens effects and confounding diseases so that our SE × MD interaction results can likely be solely explained by retinal differences associated with ametropia, such as differences in nerve fiber anatomy. We demonstrate that with increasing severity of VF loss and degree of ametropia, myopes develop more profound paracentral and nasal step VF depressions, while hyperopes develop more depression in the Bjerrum and temporal VF points (Figure 4B and Figure 5B). Furthermore, while myopia alone is associated with decreased sensitivity in the temporal sector and increased sensitivity in the nasal step sector, the pattern reverses when the interactive effect of ametropia and VF loss severity is examined. These distinct patterns indicate that different mechanisms are responsible for the effects of ametropia and glaucoma on VF loss.

We performed additional analyses after excluding subjects older than 80 years or younger than 18 years, as these patients might have a higher ratio of non-excluded pseudophakia (see Discussion) or not have age-matched controls, respectively. Similar effects on VF patterns were observed with or without the age exclusion criteria (Appendix A). Furthermore, we performed analyses after excluding eyes with MD < −18 dB, because our dataset and others indicate that the pattern standard deviation and PD values begin to normalize at this degree of VF loss severity [30]. Again, similar effects were observed with this exclusion criterion (Appendix A), indicating that our results are not caused by potential non-linearities of the PD values. Finally, we re-analyzed the data after excluding eyes with lower absolute refractive error (−1.5 D ≤ SE ≤ +1.0 D) to exclude the vast majority of pseudophakics (see Discussion). Similar results were obtained with this exclusion criterion as well (Appendix A).

Given our findings, we further examined the effect of ametropia on light sensitivity at the 4 most central VF locations on the 24-2 plot representing macular function. This experimental design was inspired by our previous work [17] showing that CRVT nasalization was significantly correlated with VF depression only in the central sector (as defined by the annular scheme [17] and the Garway-Heath scheme [31]). In the current study, we demonstrate that myopia is significantly associated with VF loss in the central four locations and that the correlation becomes progressively stronger with increasing VF loss severity (Table 1). These results are consistent with our previous finding that myopes have more nasally located CRVTs [15], which in turn is associated with deeper central VF depression [16,17]. Although we cannot conclude any causal relationships, CRVT nasalization may explain the increased susceptibility of myopic eyes to central VF loss. We and others have speculated that CRVTs can act as stabilizing forces against glaucomatous deformation of the lamina cribrosa [17]. More nasally located CRVTs in myopic eyes can result in greater mechanical strain in the temporal area, making the macular region more susceptible to glaucomatous damage.

Our finding that myopic individuals are predisposed to central vision loss is consistent with previous studies showing an association between myopia and paracentral scotomas [19,20,21,22]. Mayama et al. focusing on the central 12 points on HFA 30-2 VFs of 313 glaucoma patients, reported that myopia is associated with damage in the lower cecocentral VF [19]. Myopia was also found to be a risk factor for VF progression in the upper paracentral subfield in 92 normal-tension glaucoma patients [20]. In a recent study, Dias et al. found myopia to be associated with the presence of parafoveal scotomas in 130 glaucomatous eyes with disc hemorrhage [21]. The current study agrees with these prior works and significantly expands upon them by analyzing a dataset of over 120,000 VFs pointwise, rather than focusing only on the presence of paracentral scotomas or a subset of VF locations. Using this systematic approach, we show that myopic VF depression not only affects the cecocentral and paracentral areas but also extends to the nasal step locations, forming an arcuate pattern that corresponds to the superior and inferior arcuate bundle trajectories. Our results support the recommendations from previous studies that myopic individuals, particularly those with high myopia, deserve closer monitoring for central field defects which are highly correlated with quality of life [32,33].

While our study does not provide direct mechanistic evidence, we briefly discuss potential physiologic explanations for the VF patterns observed. First, the effect of SE alone on VF light sensitivity is likely due to structural differences between myopic and hyperopic eyes. Myopia is associated with increased axial length, optic disc tilt, and torsion [8,9,12,15,34]. Furthermore, structural parameters such as optic disc torsion [35] and abrupt change in scleral curvature [23] have been associated with VF defects in myopic eyes. The stretching or bending of optic nerve fibers due to mechanical tension may explain the increased susceptibility of myopic eyes to develop VF loss at certain locations. On the other hand, the VF patterns seen from the interaction of ametropia and glaucoma suggest that differences in RNF anatomy are responsible for the effect. The major superotemporal and inferotemporal RNF bundles, i.e., the arcuate fibers, are particularly susceptible to damage and are preferentially lost in glaucoma [36,37]. These bundles lie closer to the fovea in myopic eyes, as shown as a schematic in Figure 1B [10,11,12,13]. The reciprocal patterns observed, in which myopic VF loss shifts centrally and nasally with worsening glaucoma and hyperopic VF loss occurs in the opposite direction, correspond well to the respective locations of the arcuate fiber trajectories in myopic and hyperopic eyes.

In the present study, ametropia was significantly correlated to VF loss severity, but the effect was weak (*r* = 0.045). This likely represents a clinically insignificant result, in line with our previous study of a smaller population (*n* = 438) showing no significant association between SE and MD [15]. Our recent studies have shown, however, that optic nerve related parameters associated with myopia have specific impacts on the abnormality patterns of RNFL thickness measured by optical coherence tomography (OCT) [13,38]. Consistent with these findings, the current study shows differences in the patterns of relative light sensitivity between myopes and hyperopes, resulting in an effect modification of glaucomatous VF loss. Indeed, the magnitude of ametropia’s effect on individual PD values was small, and we would not expect ametropia alone to produce VF loss that mimics glaucoma. However, the overarching patterns of VF loss indicate that there are distinct zones of vulnerability in myopic and hyperopic eyes predisposing them to different patterns of VF loss. These patterns are exemplified in Figure 6 showing VF progression in myopic and hyperopic glaucoma patients.

There are several strengths of our study: first, we used a large sample size of over 120,000 eyes collected from multiple institutions to study the effect of ametropia on visual function. Second, we used a systematic and quantitative approach to examine the effect of the full spectrum of SE pointwise over the entire 24-2 VF, quantifying the effect of SE on PD values at each VF location. Finally, our study focuses on the interaction term (SE × MD) in the regression analysis, separating the effect of ametropia from that of VF loss severity. This specific study design addresses well-known challenges related to research on myopia and posterior eye diseases, as it filters out the various potential impacts of refractive error on the VF and allows to extract only those effects that are immediately relevant for glaucoma.

This study also has several limitations. First, because of the retrospective, cross-sectional nature of our study, we could establish associations between ametropia and VF patterns, but not causal relationships. Second, because axial length was not recorded by the HFA device, we used SE as an alternative. Third, in the absence of diagnostic information in our dataset, patients with lens related conditions could not be excluded from analysis. This is of particular relevance for cases of pseudophakia due to cataract surgery, which is a confounder when investigating impacts of ametropia on posterior eye diseases. We addressed this potential problem in two ways. First, as the prevalence of cataracts strongly increases with age, in a supplemental analysis we recalculated our results with subjects older than 80 years excluded. Second, in another supplemental analysis, we excluded all eyes with relatively mild refractive errors (−1.5 D ≤ SE ≤ +1.0 D), which is a range into which the vast majority of eyes fall after cataract surgery [39]. For either of these two additional data analyses, the effects we found were similar to the original results and did not change any of our conclusions. Apart from pseudophakia, we would like to note again that our focus on the SE×MD interaction term would extract most lens related properties as those are likely additive to VF loss patterns but would not be expected to interact with glaucomatous VF loss severity. Fourth, this study was restricted to the 24-2 pattern test. The 10-2 VF test has higher sensitivity for central vision compared to the 24-2 [40] and may be better suited to examine the detailed pattern of central VF loss. Fifth, a linear association between SE and PD was observed at most, but not all, of the VF locations (Figure 3). Nonlinear regression may be used in future studies to better model the association. Finally, the current study focused on functional data (VFs) and not structural data (e.g., RNFL defects on OCT), so we could only speculate the anatomical basis of the observed VF patterns. Future work will focus on characterizing the structure-function relationship and the effect of ametropia on this relationship.

In conclusion, utilizing a large dataset of over 120,000 VFs, we characterized the effect of ametropia on the spatial pattern of VF loss as a function of glaucoma severity. We demonstrate that myopic and hyperopic individuals are predisposed to developing different patterns of VF loss. With worsening VF loss severity, individuals with myopia have increased depressions at the paracentral and nasal step regions; conversely, hyperopes have increased depressions in the Bjerrum and temporal regions. Clinicians should be aware of these effects from ametropia and take them into account when interpreting VF loss, particularly in patients with severe VF depression. 

## Figures and Tables

**Figure 1 jcm-10-02796-f001:**
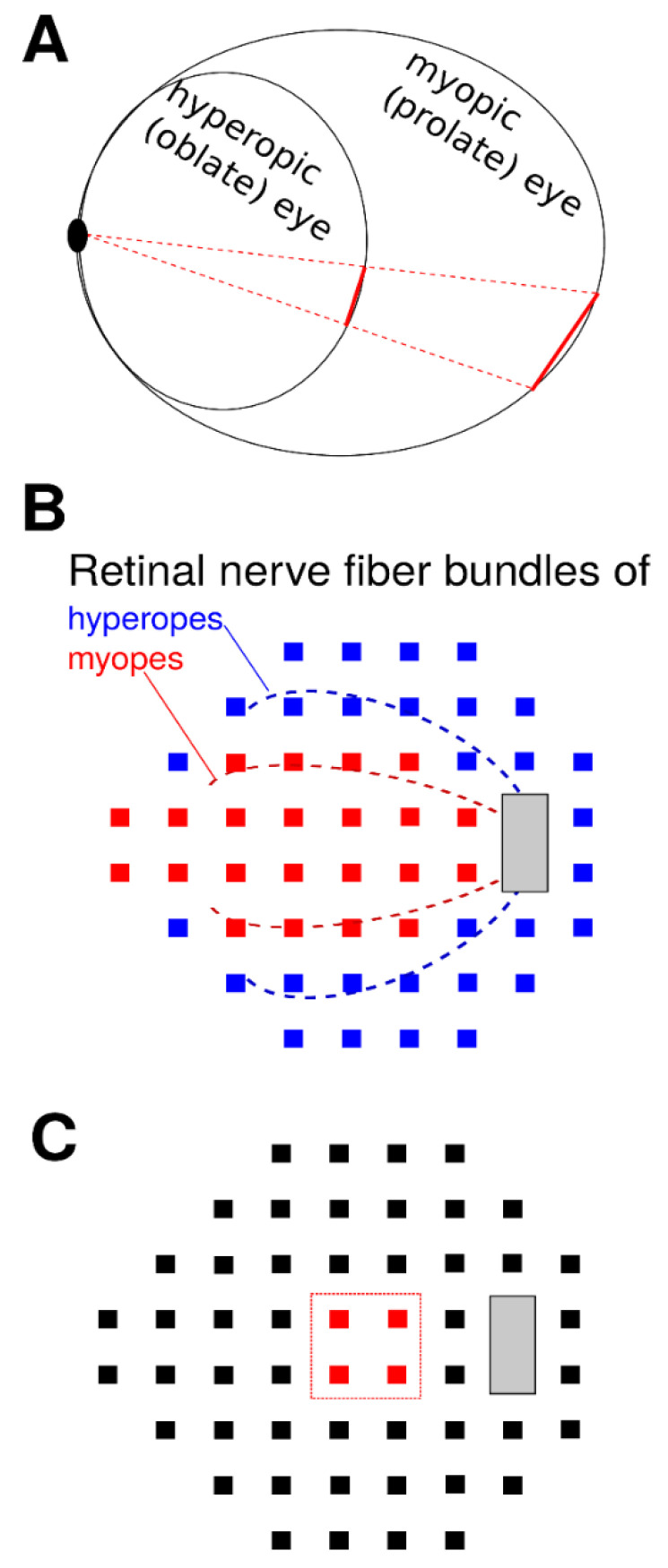
Schematic illustration of the three main hypotheses. (**A**) Ametropia is related to differences in eye length and shape, e.g., myopic eyes are longer, “curvier” (more prolate), and less regular. Therefore, we hypothesize relative differences of light sensitivity related to ametropia independent of glaucoma. (**B**) The two major retinal nerve fiber bundles, illustrated by dashed lines superimposed on the locations of a Humphrey 24-2 visual field (VF), are closer to the fovea for myopes (red lines) than for hyperopes (blue lines). Therefore, we hypothesize a center-periphery interaction effect between glaucoma severity and ametropia, schematically illustrated by the two different colors of the VF locations. (**C**) Myopia is correlated to a nasalization of the central retinal vessel trunk which, in turn, is related to glaucomatous central VF loss on the four central locations of the Humphrey 24-2 VF, illustrated in red. Therefore, we hypothesize deeper central VF depression for myopes, particularly for higher glaucoma severity.

**Figure 2 jcm-10-02796-f002:**
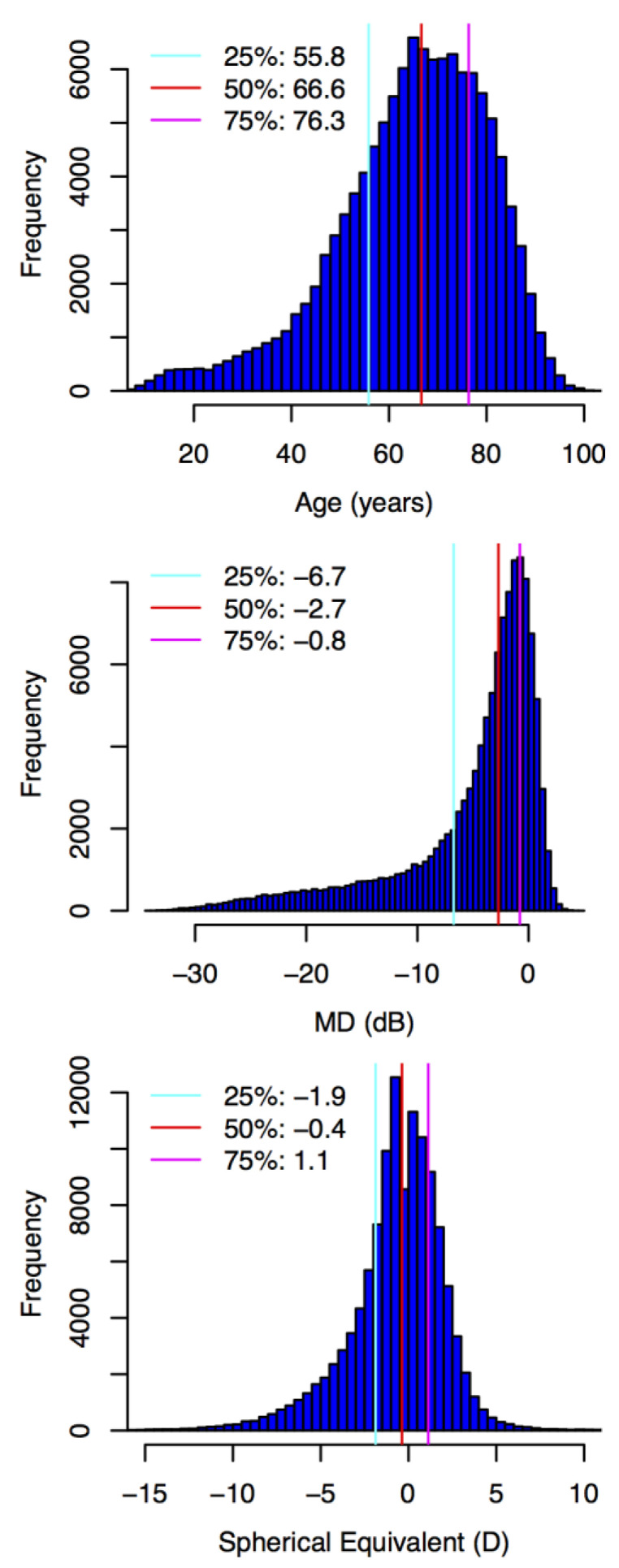
Demographic histograms of age, visual field mean deviation (MD), and spherical equivalent of refractive error (from top to bottom). Quartiles are denoted by vertical lines.

**Figure 3 jcm-10-02796-f003:**
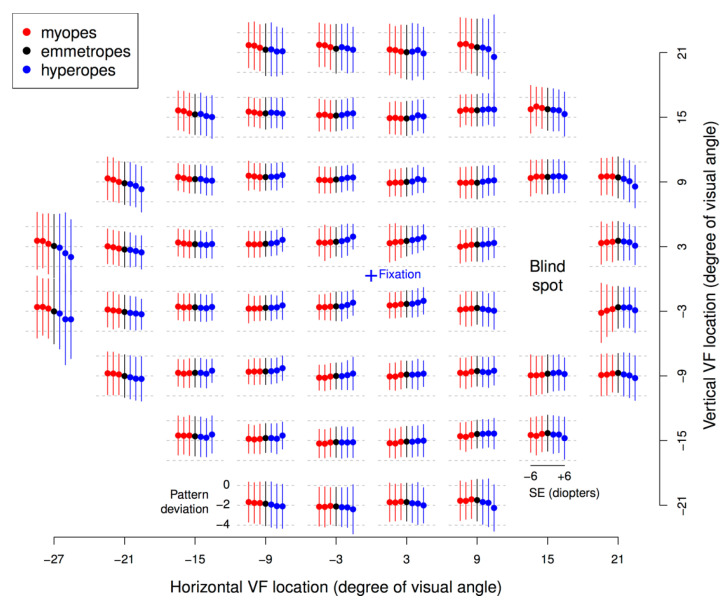
Mean pattern deviations (PD), illustrated by filled circles, and corresponding standard deviations (whiskers) grouped by bins of spherical equivalent (SE) of refractive error (bin centers: −6, −4, −2, 0, 2, 4, and 6 Diopters) for each visual field (VF) location for patients with VF mean deviations within ±1 dB. Each SE bin contains SEs within ±1 Diopter of the respective bin center. The location of fixation is denoted by the central blue cross. The two VF locations closest to the blind spot are omitted.

**Figure 4 jcm-10-02796-f004:**
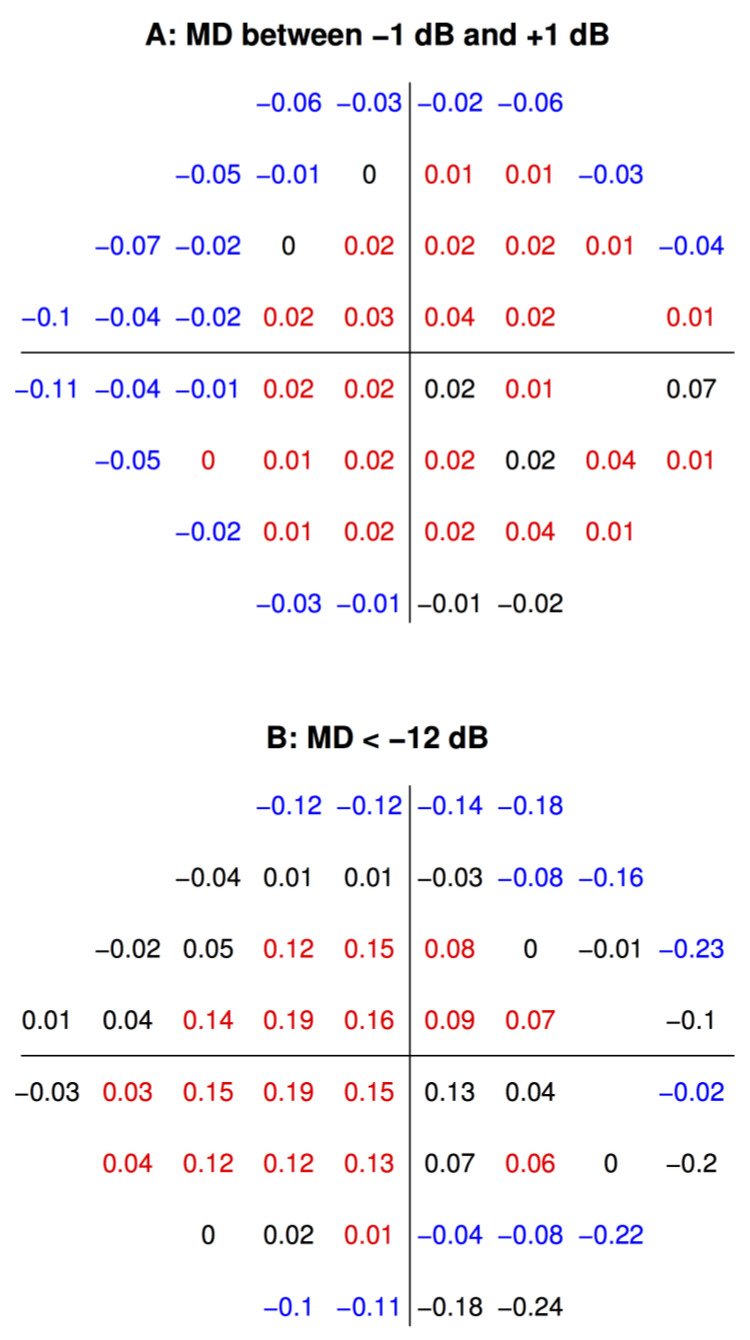
Spherical equivalent regression coefficients of pattern deviations at each visual field (VF) location for (**A**) patients with (at most) minor VF depression (mean deviation (MD) within ±1 dB) vs. (**B**) patients with severe VF depression (MD < −12 dB). Non-significant coefficients are colored in black, significant positive coefficients in red, and significant negative coefficients in blue. In short, at red/blue locations, myopes have more/less VF depression.

**Figure 5 jcm-10-02796-f005:**
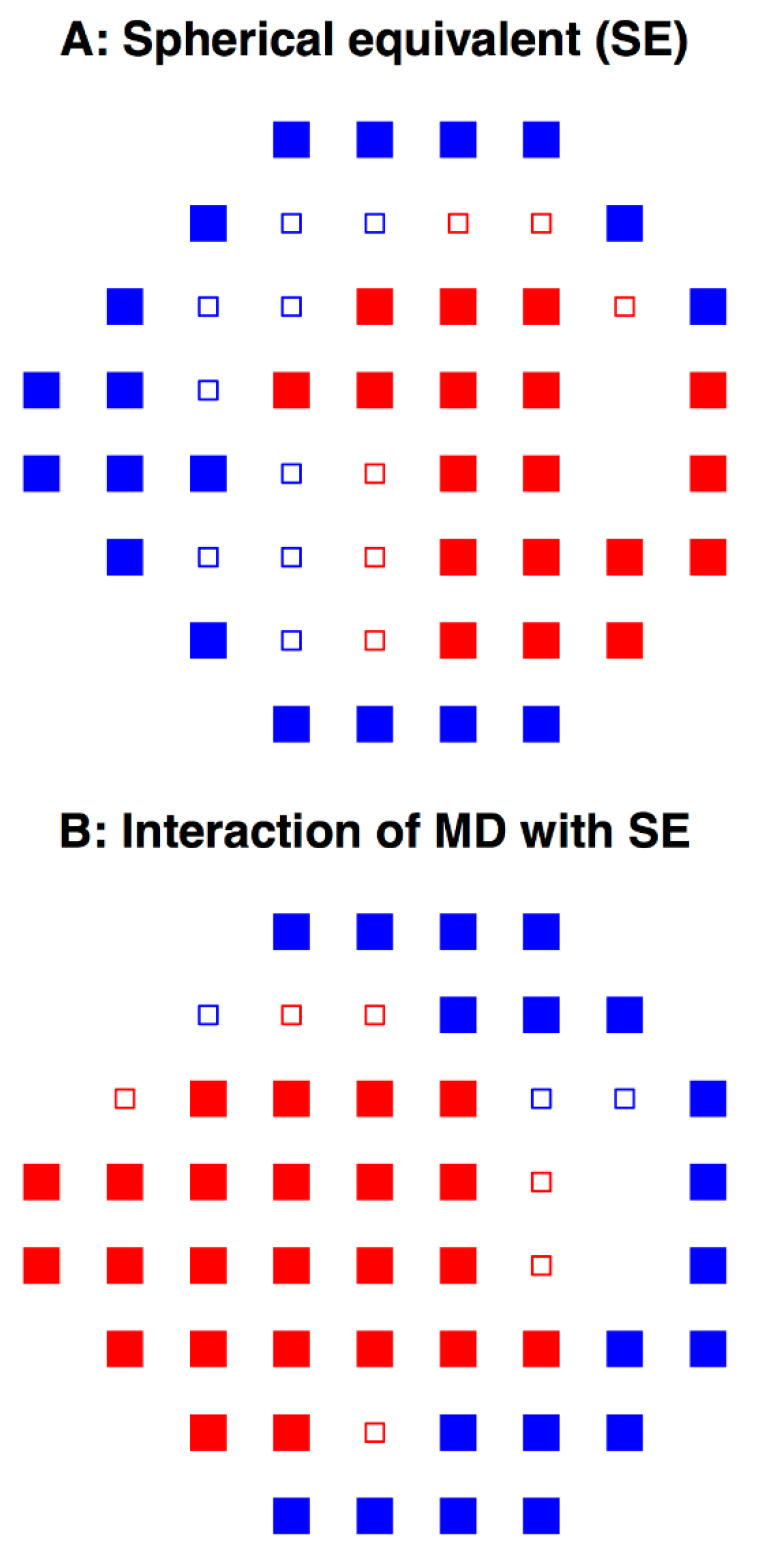
(**A**) Impact of spherical equivalent (SE) on visual field pattern deviations that are not explained by glaucoma severity (mean deviation, MD) and (**B**) interaction effects between glaucoma severity (MD) and SE. Significant locations are denoted by filled squares, non-significant locations by small, open squares. In label (**A**), red/blue locations denote positive/negative coefficients, i.e., locations where myopes have more/less VF depression regardless of glaucoma severity. In label (**B**), red/blue locations denote negative/positive coefficients of the interaction term (SE × MD). In short, at red/blue locations, increasing glaucoma severity is related to more/less VF depression in myopes.

**Figure 6 jcm-10-02796-f006:**
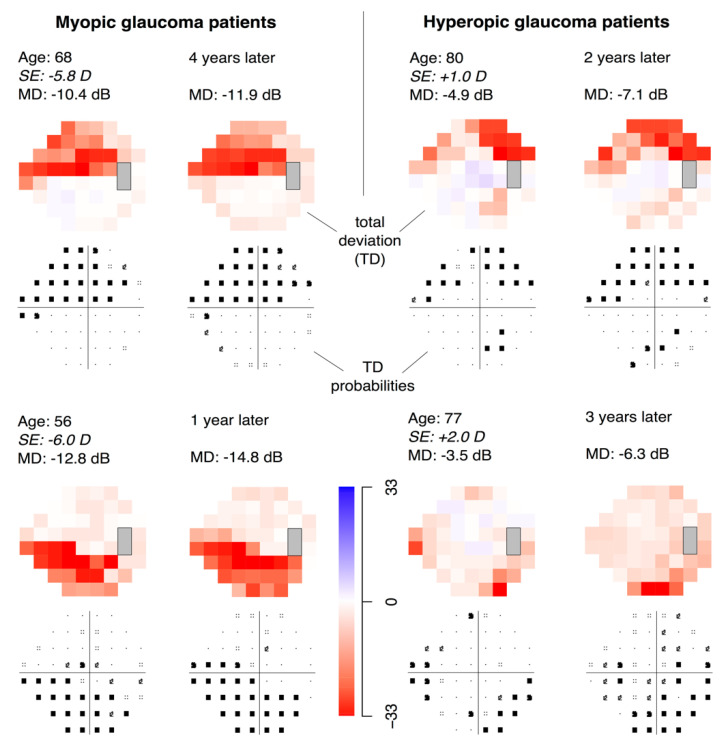
Example visual fields (VFs) of myopic and hyperopic patients with glaucoma, and the progression of VFs over time. Total deviation (TD) plots are shown for each patient: the color plots represent the numerical TD values (dB) and the grayscale plots represent the probability plots. In myopic patients (left panel), VF defects tend to be located in the paracentral and nasal step regions, whereas in hyperopic patients (right panel), VF defects tend to be located in the Bjerrum and temporal regions.

**Table 1 jcm-10-02796-t001:** Spherical equivalent regression coefficients of pattern deviations for the central four visual field (VF) locations on SITA 24-2 by VF loss severity. Each mean deviation (MD) bin contains MDs within ±3 dB of the respective bin center given in the first column. *p* values are adjusted for multiple comparisons.

MD Bin Center (dB)	SE Coefficient	*p* Value
0	0.03	6.82 × 10^−46^
−6	0.05	3.22 × 10^−14^
−12	0.06	0.000455
−18	0.12	7.04 × 10^−5^
−24	0.20	5.92 × 10^−6^

## Data Availability

The datasets and analysis programs are available from the corresponding author.

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
