# Peer review of "The Effect of Ametropia on Glaucomatous Visual Field Loss"

_jcm, 2021, doi:10.3390/jcm10132796_

Round 1

Reviewer 1 Report

Very nice paper. The strength of the paper the number of visual field analyzed, which allows the author to tease out subtle differences at each point of the visual field for both myopic and hyperopic patients. 

  1. My understanding is that hyperopic eyes are oblate. Emmetropic eye are spherical. Consider changing Fig 1A.
  2.  There is a typo in line 352. our instead of out
  3. My principle concern is the confounding myopic shift which can occur with cataracts and the confounding emmetropia with pseudophakia. This issue is addressed in the discussion.
  4. I believe the HVF assigns a reading add for all test takers.  Did that influence the results? 
  5. In the methodology section, discuss how data between right and left eyes were analyzed separately or were used to make a composite with the right eye visual field being used for graphical representation. 

Reviewer 2 Report

The manuscript in general was clear and well-written. Noteworthy is the large sample size and the extensive amount of VF test locations under study. 

Did you consider/take into account neuronal issues or trauma as possible cause of VF loss? Is it possible that this would influence the VF results and thereby also the analyses? Please clarify this.
